# Superlattice by charged block copolymer self-assembly

Jimin Shim [1], Frank S. Bates[2] & Timothy P. Lodge[1,2]

Charged block copolymers are of great interest due to their unique self-assembly and physicochemical properties. Understanding of the phase behavior of charged block copolymers, however, is still at a primitive stage. Here we report the discovery of an intriguing superlattice morphology from compositionally symmetric charged block copolymers, poly[(oligo(ethylene glycol) methyl ether methacrylate–co–oligo(ethylene glycol) propyl sodium sulfonate methacrylate)]–b–polystyrene (POEGMA–PS), achieved by systematic variation of the molecular structure in general, and the charge content in particular. POEGMA–PS self-assembles into a superlattice lamellar morphology, a previously unknown class of diblock nanostructures, but strikingly similar to oxygen-deficient perovskite derivatives, when the fraction of charged groups in the POEGMA block is about 5–25%. The charge fraction and the tethering of the ionic groups both play critical roles in driving the superlattice formation. This study highlights the accessibility of superlattice morphologies by introducing charges in a controlled manner.

[1] Department of Chemistry, University of Minnesota, Minneapolis, MN 55455, USA. [2] Department of Chemical Engineering & Materials Science, University of Minnesota, Minneapolis, MN 55455, USA. Correspondence and requests for materials should be addressed to F.S.B. (email: bates001@umn.edu) or to T.P.L. (email: lodge@umn.edu)

Block copolymers have attracted a great deal of attention over the past half-century, due to an ability to self-assemble into a variety of precisely controlled periodic nanostructures[1,2]. Several recent reports have been directed toward unveiling the thermodynamics and self-assembly of ion-containing block copolymer systems, including both salt-doped block copolymers[3–7], and intrinsically charged block copolymers containing both charged and neutral blocks[8–13]. Given their ion-embedded structures, such charged block copolymers with tailored charge fraction and architecture are expected to impact a broad range of applications, including nanolithography[14], energy storage and delivery devices[15], and pharmaceutics[16].

Despite these pioneering experimental and theoretical studies, understanding of the behavior of charged block copolymer systems must be considered to be at a primitive stage, relative to their neutral counterparts. Introduction of charges complicates the phase behavior due to significant changes in intermolecular interactions and the resulting free energy balance. For example, increasing the charge fraction in one block can lead to either suppression or enhancement of microphase separation, depending on the complex combination of parameters beyond the Flory–Huggins interaction parameter $\chi$, such as local charge organization, solvation of counterion, chain conformation, and dielectric constant[3,7,10,17–19]. Recently, Sing et al. demonstrated theoretically that self-assembly of charged block copolymers is dictated by a combination of entropy, ion solubility, electrostatic cohesion, and charge density, as probed by a hybrid self-consistent theory–liquid-state theory[20]. By manipulating the Coulombic cohesion of the system through the alteration of charge density and counterion properties, drastic shifts in the diblock-phase portraits were predicted, thereby anticipating potentially useful morphologies inaccessible with traditional neutral diblocks[20–23]. However, such theoretical predictions have not been experimentally elucidated yet, in part due to the difficulties in designing well-defined charged block copolymer model systems with precisely tuned charge fractions, and in part due to the broad parameter spaces to explore. Nevertheless, glimpses of the powerful role played by polymer-bound charges have been revealed by Park[24–28] and Balsara[29,30] and their coworkers, who demonstrated that the introduction of charged functional groups in either the PS or PEO blocks of poly(styrene)–b–poly(methylbutylene) (PS–PMB) and poly(styrene)–b–poly(ethylene oxide) (PS–PEO) diblock copolymers yields a variety of self-assembled morphologies, even from compositionally symmetric diblock systems.

In this work we describe the preparation of ion-containing diblock copolymers that offers control over the concentration and spatial distribution of charged moieties within the segregated nanodomain structure. Small-angle X-ray scattering (SAXS) experiments reveal the discovery of a superlattice state, located in phase space between the classical disordered and periodic lamellar morphologies as a function of the concentration of charge. This finding points to fascinating and versatile approaches for manipulating the nanoscale structure and properties of soft materials.

## Results

**Preparation of charged block copolymers**. As a model system of compositionally symmetric charged block copolymers, a series of poly[(oligo(ethylene glycol) methyl ether methacrylate–co–oligo(ethylene glycol) propyl sodium sulfonate methacrylate)]–b–polystyrene designated as POEGMA#–PS was synthesized by reversible addition–fragmentation chain transfer (RAFT) polymerization followed by post modification as presented in Fig. 1, where POEGMA and PS are the charged and neutral blocks, respectively. The number # in the POEGMA#–PS diblock indicates the mol% of charged POEGMA moieties along the entire POEGMA block. To precisely tune the charge fraction in the

POEGMA block, statistical copolymerization of poly(ethylene glycol) methyl ether methacrylate and poly(ethylene glycol) methacrylate was employed, where the density of hydroxyl groups along the backbone can be simply varied by adjusting the feed ratio of the two monomers. Given the comparable reactivity ratios (0.90 and 1.1) as shown in Supplementary Fig. 1, the pendant hydroxyl functionalities should be randomly distributed[31]. The resulting POEGMA was used as a macro chain transfer agent for the sequential RAFT polymerization of styrene, followed by anchoring the sodium sulfonate (–SO$_3$Na) groups to the hydroxyl groups in the POEGMA block. Details of the polymerization kinetics, end-group substitution, and $^1$H NMR spectroscopic and SEC characterization of POEGMA#–PS are presented in Supplementary Figs. 2–4, and the molecular characteristics of five POEGMA#–PS diblocks are listed in Table 1.

**Structural characteristics of charged block copolymers**. As shown by the SAXS profiles in Fig. 2, neutral POEGMA0–PS and lightly charged POEGMA3–PS exhibit a single broad peak without higher-order reflections, indicative of a disordered state. The uncharged result is consistent with the expectation for a compositionally symmetric PS–PEO diblock copolymer with comparable overall molecular weight, due to the relatively small $\chi$[32]. Attachment of 7 mol% sodium sulfonate groups to the POEGMA block, however, leads to microstructural ordering, presumably due to the significant increase in the effective segregation strength between the POEGMA# and PS blocks[33]. Strikingly, both POEGMA7–PS and POEGMA23–PS display two discernible sets of multiple reflections (denoted by unfilled and filled upside-down triangle symbols), but with distinct peak widths; this difference in peak widths between each family of peaks is more pronounced in POEGMA23–PS. Increasing the charge density further to POEGMA36–PS transforms the SAXS pattern to a set of reflections consistent with a classical lamellar morphology, as is typically found with volumetrically symmetric blocks. Note that across this series of five copolymers, the total degree of polymerization and the block compositions remain essentially constant. These unanticipated findings raise a series of questions about the nature of the intermediate morphology between disorder and lamellae, along with the underlying driving force for such unexpected ordering in this family of charged diblocks.

In order to confirm the interpretation of a lamellar morphology in POEGMA36–PS, we attempted transmission electron microscopy (TEM) experiments on cryo-microtomed thin sections (ca. 70 nm) of the material after vapor staining with RuO$_4$ using a 0.5 wt% aqueous solution. A representative TEM image of POEGMA36–PS is shown as an inset in Fig. 2, where the periodic length scale $d = 12$ nm quantitatively matches $2\pi/q^*$ obtained from the SAXS pattern. We note, however, that the layered morphology apparent in the TEM image contains rather wavy domains (see also Supplementary Fig. 5), which is unexpected based on the intense and sharp peaks found in the SAXS pattern. We believe that this is a consequence of the hygroscopic nature of the material[34]. During staining, the hydrophilic domains in the thin section of a polymer almost certainly swell with water, with subsequent drying in the electron microscope. This process likely results in significant distortion of the morphology consistent with the lack of long-range registration in the TEM image. TEM pictures obtained from POEGMA7–PS and POEGMA23–PS were also clearly inconsistent with the SAXS results, as they showed essentially no long-range order (see Supplementary Fig. 6), which we interpret as an irreversible disruption of the morphology due to swelling.

**Superlattice formation**. To better expose the intriguing self-assembly of POEGMA7–PS and POEGMA23–PS, the scattering

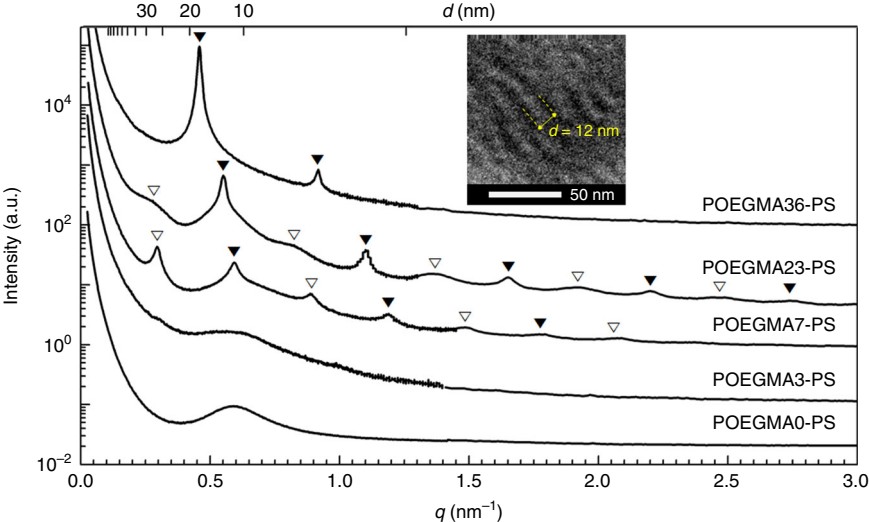

**Fig. 1** Preparation of charged block copolymers. Synthetic route to POEGMA#–PS by sequential RAFT polymerization followed by post modification, where # indicates the mol% of sodium sulfonate (–SO$_3$Na) containing POEGMA moieties in the POEGMA block

### Table 1 Molecular characteristics of POEGMA#–PS

| Sample | $M_n$ (kDa)[a] | $Đ$[b] | $N_{POEGMA-CH_3}$[c] | $N_{POEGMA-SO_3Na}$[d] | $N_{PS}$[e] | $f_{POEGMA}$[f] |
|---|---|---|---|---|---|---|
| POEGMA0–PS | 9.3 | 1.09 | 17 | 0 | 40 | 0.54 |
| POEGMA3–PS | 9.2 | 1.09 | 16 | 0.48 | 40 | 0.53 |
| POEGMA7–PS | 9.4 | 1.12 | 16 | 1.2 | 39 | 0.56 |
| POEGMA23–PS | 10 | 1.18 | 13 | 3.9 | 42 | 0.56 |
| POEGMA36–PS | 10 | 1.19 | 10 | 5.8 | 41 | 0.56 |

[a]Number average molecular weights determined by $^1$H-NMR spectroscopy. [b]Dispersities obtained by SEC in dimethylformamide (DMF) containing 0.05 M LiBr as the eluent. Number-average degree of polymerization of [c]POEGMA terminated by –CH$_3$ (x in Fig. 1), [d]POEGMA terminated by –SO$_3$Na (y in Fig. 1), and [e]PS (z in Fig. 1) determined by $^1$H-NMR spectroscopy. [f]Volume fractions of POEGMA block determined by $^1$H-NMR spectroscopy

**Fig. 2** Self-assembly of charged block copolymers. Small-angle and mid-angle X-ray scattering (SAXS and MAXS) combined profiles of the series of POEGMA#–PS at 80 °C. The filled upside-down triangle symbols indicate $q/q^*$ at multiples of 1, 2, 3, 4, 5,..., while the unfilled upside-down triangle symbols identify $q/(q^*/2)$ in multiples of 1, 3, 5, 7, and 9. The curves have been vertically shifted for clarity (inset: TEM image of POEGMA36–PS)

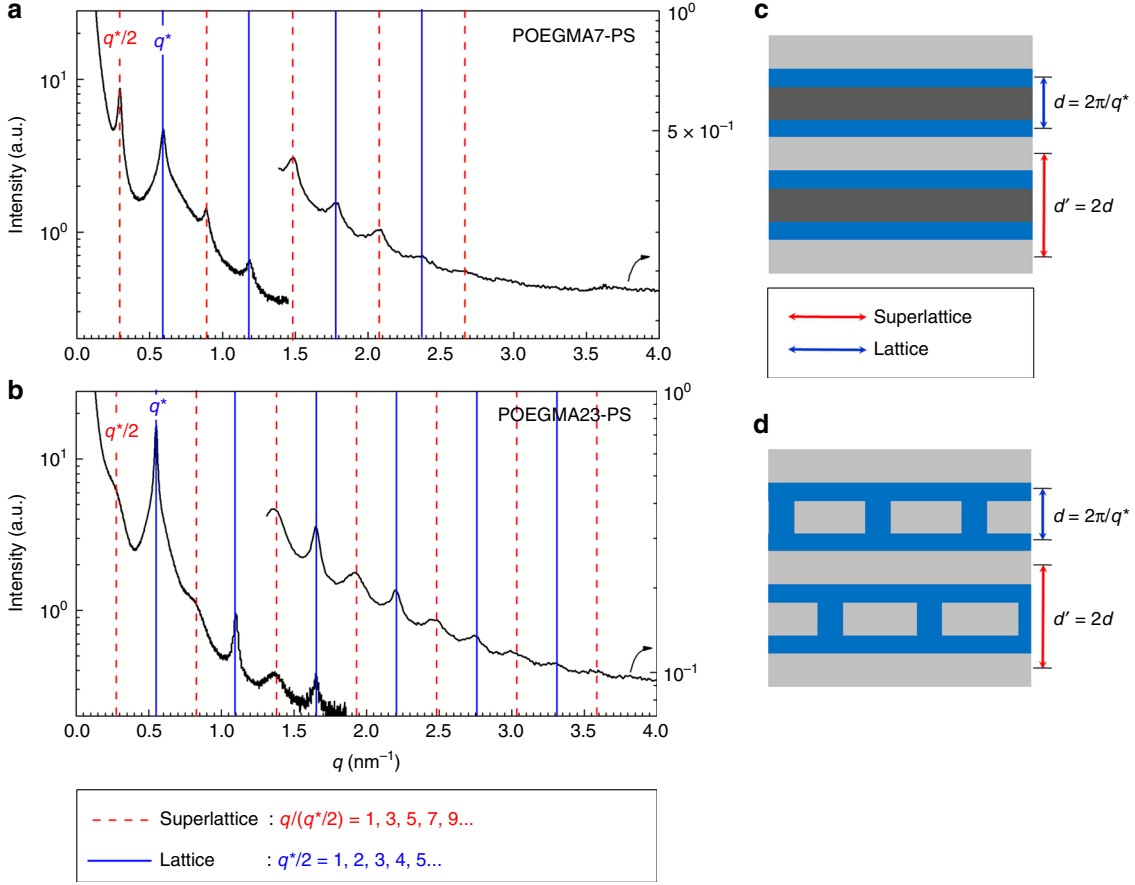

**Fig. 3** Superlattice structures. SAXS and MAXS combined profiles of **a** POEGMA7–PS and **b** POEGMA23–PS at 80 °C. The MAXS curves have been shifted vertically for clarity. **c** A representative cartoon schematic of superlattice morphology. **d** One plausible example of superlattice morphology

traces have been plotted separately in Fig. 3a, b, where the red-dashed and blue vertical lines indicate the Bragg peak positions corresponding to each distinct set of reflections. The diffraction peaks associated with the blue lines indicate $q/q^*$ multiples of 1, 2, 3, 4, 5 and 6, while those shown by the red-dashed lines are precisely coincident with the sequence in $q/(q^*/2)$ of 1, 3, 5, 7, 9, 11 and 13. Both sets of reflections are consistent with a layered structure with a high degree of long-range order. Referring to Fig. 2, the $q/q^*$ sequence of reflections (blue vertical lines) is clearly correlated with an increasing periodic length scale, as the average number of charges per POEGMA block increases, bracketed by disorder and simple lamellae. This pattern is consistent with the expectation $d \sim \chi^{1/6}$ (see ref. [35]), since increasing the ion content drives up the effective value of $\chi$[33]. Hence, we refer to $q^*$ as the primary peak position for each series. Remarkably, the alternate sets of Bragg peaks (red-dashed lines) are positioned at exactly twice the lattice spacing associated with $q^*$, indicative of period doubling that is directly coupled to the principal layer spacing. Based on these results, we speculate that the morphology is a lamellar superlattice structure with an …ABA'BAB… sequence of domains as depicted qualitatively in Fig. 3c. One plausible example of morphological arrangement that would yield an array of Bragg peak positions consistent with the scattering traces is also suggested in Fig. 3d, which shows alternating stacking of continuous and perforated layers. To the best of our knowledge, this kind of scattering pattern has never been reported for any other block copolymer system. However, this type of scattering pattern bears a striking resemblance to that obtained from certain oxygen-deficient perovskite derivatives such as Brownmillerite, which features distinctive sets of full- and half-order Bragg reflections originating from the doubling of the unit cell due to alternate stacking of octahedral and tetrahedral layers associated with the oxygen vacancies in the tetrahedral layers[36,37]. The results in Fig. 3b from POEGMA23–PS are especially noteworthy with regard to the sharp versus broad reflections attributed to the lattice and superlattice, respectively, a feature that is also found in certain perovskites with 50% oxygen deficiency.

Although the underlying origin of superlattice formation remains to be fully elucidated, we speculate that it is associated with a competition among several factors beyond a simple, effective $\chi$ parameter. The inherent conformational asymmetry in this system (i.e., higher self-concentration in the POEGMA vs. PS domains) accompanied by the sparse placement of charges at the terminus of the branches should lead to a state of considerable packing frustration[38,39]. This effect will compete with strong electrostatic correlations between charges, where the Bjerrum length ($l_B$)[10] lies in the range of 6–11 nm (see Supplementary Fig. 7). Interestingly, this is shorter than, but on the same order of magnitude as, the superlattice length scale. This raises the possibility that the formation of a superlattice reflects the mismatch between the length scales that optimize space filling (driven by composition) and electrostatic correlations. Consistent with this, complex interplay between domain interfacial tension and block stretching (the classical factors that compete in determining the microdomain size and geometry) and packing frustration and electrostatic interactions collapses when the charge fraction is increased to 36 mol%. This suggests that the average spacing between charged groups, estimated in POEGMA7–PS and POEGMA23–PS to be 2.2 nm

and 1.2 nm, respectively (see Supplementary Fig. 7), represents a key factor in driving formation of a superlattice. Furthermore, any net local attraction between charges and their counterions, such as those that lead to clustering in ionomers, and which could be facilitated by the conformational freedom of ions tethered at the ends of flexible POEGMA side chains, would give rise to a larger length scale, on the same order of magnitude as both the domain spacing and Bjerrum length. These phenomena are at least broadly consistent with the important roles of ionic correlations and dielectric mismatch anticipated by theory, although a superlattice has yet to be predicted[20,23]. Furthermore, dispersity in the number of charges per chain could also play a significant role in superlattice formation. Since the POEGMA#–PS system possesses a relatively sparse distribution of charges along the low-molecular-weight POEGMA block, there might be inherent inhomogeneities in the charge distribution per each block copolymer, inducing a nontrivial phase separation between uncharged and charged species[19], which could contribute to the superlattice formation. A simple analysis (see Supplementary Fig. 8) suggests that for POEGMA7–PS, about one-third of the POEGMA blocks could have no charged groups at all. However, for POEGMA23–PS, the fraction of uncharged chains drops below 3%, indicating that phase separation of charged and uncharged chains cannot be the primary driving force for superlattice formation.

**Effect of constrained ionic species on superlattice formation**. In order to isolate the consequences of localizing the charged moieties on the branched POEGMA blocks versus concentration of charged groups, we doped POEGMA0–PS with NaCF$_3$SO$_3$ salt. This relieves the constraints associated with tethering the ionic moieties to the POEGMA blocks, while maintaining the same average ion concentration. Figure 4a illustrates these two approaches to introduce charge into the diblock: (i) attachment to the backbone and (ii) extrinsically doping salt into the neutral POEGMA0–PS. SAXS data obtained from mixtures of POEGMA0–PS and NaCF$_3$SO$_3$ prepared with the same ion concentration [Na$^+$]/[EO] as the corresponding POEGMA#–PS counterparts are compared with the compositionally equivalent POEGMA#–PS data in Fig. 4b. NaCF$_3$SO$_3$ partitions almost exclusively to the more polar POEGMA block due to the large solvation energy gained from the association of POEGMA and salt[3]. Although a difference in anion size relative to the –SO$_3$ moieties in POEGMA#–PS could affect the electrostatic interaction strength, the overall effect on the enthalpic component of the effective $\chi$ parameter is assumed to be roughly equivalent in both cases. Free salt also leads to ordering of POEGMA0–PS, demonstrated by intense and narrow principal Bragg peaks, but there is no evidence of a superlattice at any salt concentration. Moreover, there are considerably fewer and weaker-intensity higher-order reflections, indicative of decreased long-range order for the case of unbound charges. Addition of the unconstrained ions also produces a considerably larger domain period, ~50% greater for [Na$^+$]/[EO] = 0.048. At the highest free salt content [Na$^+$]/[EO] = 0.072, the counterpart for POEGMA36–PS, we find two sets of Bragg peaks indicative of a mixture of hexagonal and lamellar morphologies, as opposed to pure lamellae[24,40]. These qualitative and quantitative differences demonstrate the profound consequences of coupling ionic species to the block architecture.

**Discussion**

We have demonstrated that tethering a sparse number of ionic species to the termini of the branched POEGMA units in a compositionally symmetric POEGMA–PS diblock copolymer leads to the formation of a superlattice morphology. Reducing or increasing the number of charged groups leads to disordered and lamellar morphologies, respectively. Substitution of free salt for

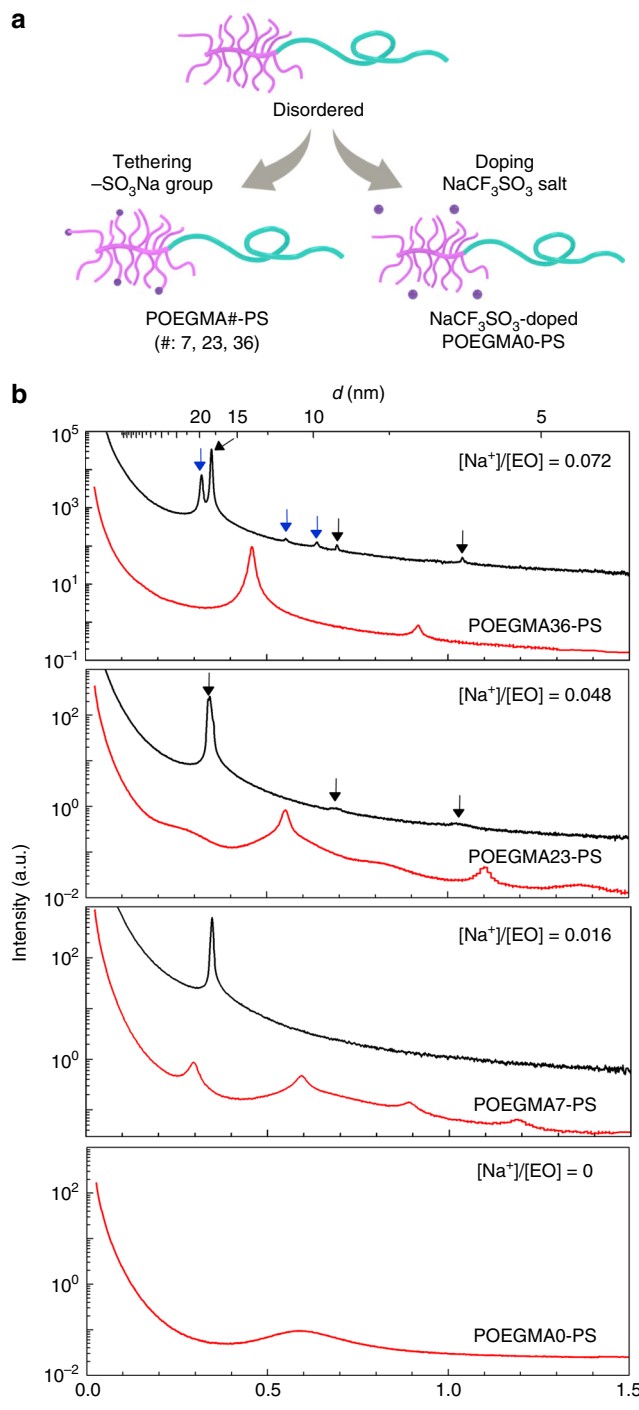

**Fig. 4** Effects of constraints of ionic species on self-assembly. **a** Two ways of introducing ions in this study. **b** SAXS traces of POEGMA#–PS as indicated by the red lines and NaCF$_3$SO$_3$-doped POEGMA0–PS ([Na$^+$]/[EO] = 0.016, 0.048, and 0.072) as indicated by the black lines (temperature: 80 °C), where EO represents ethylene oxide. The black arrows indicate $q/q^*$ at multiples of 1, 2, and 3, while the blue arrows identify $q/q^*$ at multiples of 1, $3^{1/2}$, and $4^{1/2}$, where $q^*$ is the primary peak position for each morphology

the tethered ionic species at the same average concentration eliminates the superlattice state, leading to a more traditional sequence of diblock copolymer states: disordered, lamellar, and mixed lamellar/hexagonal phases with increasing salt content. The latter results are consistent with elevating the classical

Flory–Huggins interaction parameter due to localization of the salt in the POEGMA microdomain spaces. Tethering charged groups to the POEGMA block introduces a competition between electrostatic interactions subject to the frustration associated with the highly self-concentrated POEGMA blocks. This preliminary set of results points to a powerful platform for designing nanostructured soft materials that contain charged moieties.

## Methods

**Synthesis of macro chain transfer agent.** The following representative synthetic procedure is for a macro chain transfer agent for POEGMA23–PS. Poly[(oligo (ethylene glycol) methyl ether methacrylate–co–oligo(ethylene glycol) methacrylate)] (POEGMA) was synthesized by reversible addition–fragmentation chain transfer (RAFT) polymerization. Oligo(ethylene glycol) methyl ether methacrylate (15.0 g, 50.0 mmol), oligo(ethylene glycol) methacrylate (5.38 g, 14.9 mmol), 2-cyano-2-propyl benzodithioate (0.988 g, 4.46 mmol), and AIBN (0.146 g, 0.892 mmol) were dissolved in 60 mL of anhydrous tetrahydrofuran (THF) and added to a Schlenk flask equipped with a magnetic stirring bar and a condenser. The flask was sealed with a Chem-Cap valve (Chemglass) and degassed by five freeze–pump–thaw cycles. The flask was backfilled with argon gas and placed in an oil bath at 70 °C. After 24 h, the reaction was rapidly quenched by liquid nitrogen and the crude solution was precipitated in n-hexane. The dissolution–precipitation cycles were repeated three times, and the purified polymer was isolated. After being dried under dynamic vacuum, red-colored viscous liquid was obtained. $^1$H NMR (400 MHz, DMSO–$d_6$): δ 4.55 (s, 4 H), 3.89–4.27 (s, 34 H), 3.42–3.74 (m, 345 H), 3.32 (s, 39 H), and 0.79–1.81 (m, 85 H).

**Synthesis of a block copolymer.** The following representative synthetic procedure is for a block copolymer prior to post modification for POEGMA23–PS. Poly [(oligo(ethylene glycol) methyl ether methacrylate–co–oligo(ethylene glycol) methacrylate)]-b-polystyrene was synthesized by sequential RAFT polymerization using the POEGMA as a macro chain transfer agent. In total, 20.0 g of POEGMA, styrene (33.7 g, 32.4 mmol), and AIBN (0.124 g, 0.753 mmol) were dissolved in 60 mL of THF and added to a Schlenk flask equipped with a magnetic stirring bar and a condenser. The flask was subjected to three freeze–pump–thaw cycles, and polymerization was conducted in an oil bath at 80 °C for 38 h. The reaction was quenched by liquid nitrogen followed by precipitation in n-hexane three times. After being dried under dynamic vacuum overnight, pink-colored powder was obtained. The obtained polymer (30.0 g, 3.00 mmol of benzodithioate end group) and AIBN (9.85 g, 60.0 mmol) were dissolved in 100 mL of THF and added to a two-neck round-bottomed flask equipped with a magnetic stirring bar and a condenser. Argon gas was purged through the flask for 30 min, and the flask was placed in an oil bath at 80 °C and stirred for 8 h. The crude solution was precipitated in n-hexane three times. After being dried under dynamic vacuum overnight, white powder was obtained. $^1$H NMR (400 MHz, DMSO–$d_6$): δ 6.34–7.38 (m, 210 H), 4.55 (s, 4 H), 3.89–4.27 (s, 34 H), 3.42–3.74 (m, 345 H), 3.32 (s, 39 H), and 0.79–1.81 (m, 85 H).

**Synthesis of a charged block copolymer.** Poly[(oligo(ethylene glycol) methyl ether methacrylate-co-oligo(ethylene glycol) propyl sodium sulfonate methacrylate)]-b-polystyrene (POEGMA23–PS) was synthesized by the following post modification. POEGMA–PS (22.0 g, 9.30 mmol of hydroxyl group), sodium hydride (60% dispersion in mineral oil) (0.744 g, 18.6 mmol), and 100 mL of anhydrous THF were charged to a two-neck round-bottomed flask equipped with magnetic stirring bar and a condenser in an argon-filled glove box. The flask was sealed with a rubber stopper and removed from the glove box. The reaction was conducted in an oil bath at 80 °C for 24 h and 1,3-propane sultone (2.27 g, 18.6 mmol) was subsequently added by dropwise to the flask by a syringe. After 48 h, the flask was cooled to room temperature in an ice bath. The excess sodium hydride was removed from the crude solution by filtration. The obtained solution was concentrated by evaporating THF and precipitated in diethyl ether three times. The precipitated product was additionally washed with distilled water to completely remove any salt residue. After being freeze-dried under dynamic vacuum, slightly yellow powder was obtained. Note that other POEGMA#–PS with different charge fractions were also synthesized by the same protocol, except for using different amounts of chemicals. $^1$H NMR (400 MHz, DMSO–$d_6$): δ 6.34–7.38 (m, 210 H), 3.89–4.27 (s, 34 H), 3.42–3.74 (m, 345 H), 3.32 (s, 39 H), 2.33 (m, 8 H), and 0.79–1.81 (m, 85 H).

**Density determination.** The density of PS is determined to be 1.043 g cm$^{-3}$ according to the previous report[41]. The density of POEGMA block in the POEGMA#–PS was estimated by the group contribution approach of Van Krevelen[42], which yields 1.080, 1.085, 1.103, and 1.116 g cm$^{-3}$ for POEGMA3–PS, POEGMA7–PS, POEGMA23–PS, and POEGMA36–PS, respectively. The density of the POEGMA block in the NaCF$_3$SO$_3$-doped POEGMA0–PS was estimated by the equation $\rho_{POEGMA+NaCF_3SO_3} = \phi_{POEGMA}\rho_{POEGMA} + \phi_{NaCF_3SO_3}\rho_{NaCF_3SO_3}$, where $\phi$ is the relative volume fraction of each component in the mixed phase. The $\rho_{NaCF_3SO_3}$

value was estimated by the equation $\rho_{NaCF_3SO_3} = 172.06$ g mol$^{-1}$/[$N_A \times 4\pi/3$ $(R_{Na}^{+3} + R_{CF_3SO_3}^{-3})] = 3.211$ g cm$^{-3}$, where $N_A$ is Avogadro's number and $R$ is an ionic radius value.

**Determination of Bjerrum length.** The Bjerrum length ($l_B$) of the system, at which the electrostatic potential equals the thermal motion energy, was determined by the equation given by $l_B = e^2/(4\pi\varepsilon_0\varepsilon_r k_B T)$, where $e$ is the elementary charge, $\varepsilon_0$ is electric permeability of vacuum, $\varepsilon_r$ is the dielectric constant of the medium, $k_B$ is Boltzmann constant, and $T$ is the absolute temperature in Kelvin[10]. Since the $\varepsilon_r$ strongly depends on the local concentration of POEGMA and PS phases[21], the $\varepsilon_r$ value is estimated by a simple local volume fraction-weighted average, $\varepsilon_r = \varepsilon_{r, POEGMA}\phi_{POEGMA} + \varepsilon_{r, PS}\phi_{PS}$[7], where the $\varepsilon_{r, POEGMA}$ and $\varepsilon_{r, PS}$ are dielectric constants of pure POEGMA and PS phase, and $\phi_{POEGMA}$ and $\phi_{PS}$ are local volume fractions of POEGMA and PS. We have assumed that $\varepsilon_{r, POEGMA} = 7.5$ and $\varepsilon_{r, PS} = 4.0$ for our system, which are the reasonable values for poly(ethylene oxide) (PEO) [43] and PS[44].

**Synchroton small-angle X-ray scattering (SAXS) experiments.** Approximately 20 mg of each sample was freeze-dried for at least 1 week under dynamic vacuum and hermetically sealed in TZero aluminum DSC pans (TA Instruments) in an argon-filled glove box to prevent oxidative degradation of the samples during the SAXS experiments. Each DSC pan containing the sample which was prepared by solution-casting method (solvent: THF) followed by vacuum drying, was annealed at 130 °C for 9 days and immersed into liquid nitrogen to kinetically trap the equilibrium morphologies before being subjected to the beamline. Synchrotron SAXS experiments were performed at DND–CAT Sector (beamline 5–ID–D) at the Advanced Photon Source (APS) located at Argonne National Laboratory. Beam energies and sample-to-detector distances of 17 keV and 8.5 m were used. A Linkam DSC600 stage was used to control the heating and cooling cycles during the experiments. Two-dimensional scattering patterns were collected on a Rayonix CCD area detector. The resulting isotropic scattering patterns were azimuthally integrated to give the scattered intensity as a function of the magnitude of the scattering wave vector $q = 4\pi\sin(\theta/2)/\lambda$, where $\theta$ is the scattering angle and $\lambda$ ( = 0.7293 Å) is the wavelength of the incident beam. All the SAXS data in this study are presented as their original form without a background subtraction.

**Transmission electron microscopy.** TEM specimens were prepared by following the same thermal history with the SAXS experiments and rapidly quenched by liquid nitrogen to rapidly capture the equilibrium morphology. The thin polymer sections with a thickness of ~70 nm were obtained by cryo-micotoming on a Leica EM UC6 at −80 °C and collected on 400-mesh copper grids. The thin sections on the grids were stained over RuO$_4$ vapor for 40 min using a ruthenium tetroxide (RuO$_4$) 0.5% aqueous solution. TEM experiments were performed using a Tecnai G2 Spirit Biotwin TEM at 120 kV of an acceleration voltage.

## Data availability

The experimental data that support the findings of this study are available in the Data Repository for the University of Minnesota (DRUM) at https://doi.org/10.13020/qsa6-qg08[45].

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

## Acknowledgements

This work was supported by the Office of Basic Energy Science (BES) of the U.S. Department of Energy (DoE), under Contract DE-FOA-0001664. The SAXS experiments were performed at DuPont-Northwestern-Dow Collaborative Access Team (DND-CAT) 5-ID at the Advanced Photon Source, Argonne National Laboratory. The TEM experiments were carried out in the Characterization Facility, University of Minnesota, a member of the NSF-funded Materials Research Facilities Network (www.mrfn.org) via the MRSEC program.

## Author contributions

The project was conceived and supervised by F.S.B. and T.P.L. The synthetic design and experiments were performed by J.S. All authors contributed to the discussion of the results and preparation of the paper.

## Additional information

**Competing interests:** The authors declare no competing interests.

