## [Peer Review File · Nature Communications]

Reviewers' comments:

Reviewer #1 (Remarks to the Author):

This is a fascinating study. The authors synthesize neutral-charged diblock copolymers where the charge content and its distribution can be controlled. In the absence of charges, the diblock is chemically compatible; that is, it is in a disordered state. As the charge content increases, a new superlattice emerges and when the charge content increases further, a periodic lamellar microstructure is obtained. This is remarkable since it shows that Coulomb interactions induce segregation into non-trivial microstructures. That is, the effect cannot be described by an increase in the degree of incompatibility through a larger value of the classical Flory parameter. The authors show the superlattice is composed of layers which support alternating continuous and perforated lamellar with a segregated nanodomain of the order of the Bjerrum length. Cleverly, the authors show that when doping the structure to produce a similar charge content in the phase that solvates the charges, the system segregates but not to a new morphology. This is consistent with the theory of Sing et al published in Nature Materials and reference therein. The dielectric mismatch alone generates the same microphases as in non-charged systems by simply increasing the degree of incompatibility (i.e., larger Flory parameter) but when ionic correlations are included, shifting is not enough when the charged content is on one of the blocks. In this later case, the chain is capable of generating new phases, not yet described, due to the effect of strong ionic correlations. This paper shows one of all possible not yet explored nanostructures due to the competition of the many length scales that Coulomb interactions bring in organic matter including the Bjerrum length, the size of the ions and the average distance between charges. I strongly recommend the publication of this work.

Reviewer #2 (Remarks to the Author):

Shim, Bates, and Lodge provide evidence of a superlattice structure in the self-assembly of charged block copolymers. This arises due to the presence of fixed charges on a poly(oligo(ethylene glycol)) acrylate monomers that are copolymerized to form a charged A block, with a B block that is PS. The authors see scattering patterns characteristic of two lamellar length scales, which are a factor of two different in their scattering peak positions. This is attributed to an ABA'BA type lamellar structure. Curiously, they only observe this for fixed charges, whereas introduction of a salt that can complex to the EO (similar to an annealed charge along the chain) does not exhibit this behavior.

This is an interesting result, and significant in that it shows how charge may be useful to obtain new block copolymer morphologies not typically obtained with uncharged block copolymers. This is suitable for publication in Nature Communications, pending the authors considering - and addressing in the manuscript - an alternate hypothesis for the observed behavior, along with a few minor comments.

I wonder if there is another explanation for the observed behavior, related to the dispersity in the number of charges per chain. Notably, the authors consider chains with relatively small degrees of polymerization for the A block, and among these chains there are often very few charges (for the polymers in question, there are 1.2 and 3.9 charged monomers out of ca. 17 monomers on average).

I'll suggest the authors provide a prediction - perhaps back-of-the-envelope - for how many chains have no charges at all. It is plausible that the two different A vs. A' lamellae just have different populations of charges, or one has all the uncharged chains while the other has all the charged chains. This could arise one of two ways (or a combination of both), with the uncharged vs. charged blocks being sufficiently chemically different due to functionalization or perhaps there is a driving force to maximize the local charge concentration due to charge correlations. Anyway, I'll

ask that the authors comment on this alternative hypothesis, since it is also consistent with the lack of the superlattice phase in the annealed salt.

Other minor comments:

-Ln. 130 - The authors should specify that they consider χ_{eff} , or an effective χ , since the quantity they consider includes electrostatic interactions.

-Ln. 153 - I am unsure about the argument here. Yes, the Bjerrum length is long, but the presence of nearby charges (average distance $\ll l_B$ as mentioned in Ln. 163) will cause any correlations - impactful as they should be to the thermodynamics of these systems - to 'die down' over a shorter length scale. I'll recommend elaboration on this point, to clarify what the authors mean.

-Ln. 163 - Along these lines, if there are only a few charged monomers per chain, and even these short chains will interact with many neighboring other chains, in what way would they constrain the polymers be due to the electrostatic correlations? Would such an effect change the observation in Ln. 130 that $d \sim \chi^{1/6}$? I am confused by the physical argument here, so further conceptual clarification or exposition would be helpful.

-Ln. 178 - The strength of the charged interaction is due not only to the underlying partitioning, dielectric constant, and concentration, but also the size of the charge (i.e. the effective diameter of the species). This is true for both the Wang and Olvera de la Cruz predictions. While the use of a similar ion perhaps minimizes this as much as experimentally possible, the size of the pendant SO₃ versus the CF₃SO₃ may yet affect this comparison.

Reviewer #3 (Remarks to the Author):

This paper reports the synthesis and morphology of POEM-based polymers bearing charged moieties. The main finding of this work is the formation of interfacial layers from charge-containing block copolymers, which is unusual in literature. SAXS profiles in Fig. 2 are quite clear that the POEGMA7-PS and POEGMA23-PS form the superlattice. This reviewer thinks that this is an interesting paper to attract broad interests from polymer (physics, chemistry, and electrolytes) communities.

Questions remaining are (1) how those polymers can form so-called superlattice and (2) why not for POEGMA36-PS? In Fig. 3c and Fig. 3d, the authors speculated superlattice morphology. This is puzzling because of random copolymer characteristics of POEGMA, anticipated based on analogous reactivity ratios of two monomers. For POEGMA7-PS, the number of POEGMA-SO₃Na is 1.2. This implies that with nominally one POEGMA-SO₃Na unit, high electron density interfacial layers were developed to result in such sharp and multiple Bragg peaks. In fact, the scattering intensity of the peak at $q/(q^*/2) = 1$ is strong, similar to that at $q/q^* = 1$. At the same time, the peak at $q/(q^*/2) = 1$ is quite sharp, compared with relatively broad peak at $q/q^* = 1$. To this reviewer, this is not easy to imagine how and why POEGMA-SO₃Na moieties randomly connected to neutral POEGMA are self-assembled into such structures. What would be the thickness of such layers? TEM image in Fig. 2 is not much informative.

With increased number density of tethered -SO₃Na group, POEGMA23-PS having 3.9 units (that of neutral POEGMA is 13), the authors conjectured interconnected charged POEGMA phases, as shown in Fig. 3d. This reviewer believes that such speculation was made by the smeared superlattice peaks. But again, how randomly distributed charged POEGMA chains (1 per every 3-4 neutral units) can be assembled into such morphology?

One last minor point:

If the sulfonate is in different forms, i.e., -SO₃H or SO₃Li, does the sample show the same superlattice peaks? Or does it become indistinguishable because of low scattering contrast?

Comments of Reviewer 1

This is a fascinating study. The authors synthesize neutral-charged diblock copolymers where the charge content and its distribution can be controlled. In the absence of charges, the diblock is chemically compatible; that is, it is in a disordered state. As the charge content increases, a new superlattice emerges and when the charge content increases further, a periodic lamellar microstructure is obtained. This is remarkable since it shows that Coulomb interactions induce segregation into non-trivial microstructures. That is, the effect cannot be described by an increase in the degree of incompatibility through a larger value of the classical Flory parameter. The authors show the superlattice is composed of layers which support alternating continuous and perforated lamellar with a segregated nanodomain of the order of the Bjerrum length. Cleverly, the authors show that when doping the structure to produce a similar charge content in the phase that solvates the charges, the system segregates but not to a new morphology. This is consistent with the theory of Sing et al published in Nature Materials and reference therein. The dielectric mismatch alone generates the same microphases as in non-charged systems by simply increasing the degree of incompatibility (i.e., larger Flory parameter) but when ionic correlations are included, shifting is not enough when the charged content in on one of the blocks. In this later case, the chain is capable of generating new phases, not yet described, due to the effect of strong ionic correlations. This paper shows one of all possible not yet explored nanostructures due to the competition of the many length scales that Coulomb interactions bring in organic matter including the Bjerrum length, the size of the ions and the average distance between charges. I strongly recommend the publication of this work.

Answer: We appreciate the thoughtful comments and recommendation for publication.

Comments of Reviewer 2

Shim, Bates, and Lodge provide evidence of a superlattice structure in the self-assembly of charged block copolymers. This arises due to the presence of fixed charges on a poly(oligo(ethylene glycol)) acrylate monomers that are copolymerized to form a charged A block, with a B block that is PS. The authors see scattering patterns characteristic of two lamellar length scales, which are a factor of two different in their scattering peak positions. This is attributed to an ABA'BA type lamellar structure. Curiously, they only observe this for fixed charges, whereas introduction of a salt that can complex to the EO (similar to an annealed charge along the chain) does not exhibit this behavior.

This is an interesting result, and significant in that it shows how charge may be useful to obtain new block copolymer morphologies not typically obtained with uncharged block copolymers. This is suitable for publication in Nature Communications, pending the authors considering - and addressing in the manuscript - an alternate hypothesis for the observed behavior, along with a few minor comments.

I wonder if there is another explanation for the observed behavior, related to the dispersity in the number of charges per chain. Notably, the authors consider chains with relatively small degrees of polymerization for the A block, and among these chains there are often very few charges (for the polymers in question, there are 1.2 and 3.9 charged monomers out of ca. 17 monomers on average). I'll suggest the authors provide a prediction - perhaps back-of-the-envelope - for how many chains have no charges at all. It is plausible that the two different A vs. A' lamellae just have different populations of charges, or one has all the uncharged chains while the other has all the charged chains. This could arise one of two ways (or a combination of both), with the uncharged vs. charged blocks being sufficiently chemically different due to functionalization or perhaps there is a driving force to maximize the local charge concentration due to charge correlations. Anyway, I'll ask that the authors comment on this alternative hypothesis, since it is also consistent with the lack of the superlattice phase in the annealed salt.

Answer: The reviewer has raised an important point on the effect of dispersity in the number of charges per chain on the superlattice formation. To explore this hypothesis, we have estimated the component distribution probability profiles of each POEGMA#–PS system by assuming a simple binomial distribution of charged and uncharged POEGMA monomers along the backbone. This is a reasonable approach given the fact that the reactivity ratios are so close to unity. The probability that the number of charged POEGMA monomers in POEGMA block equals n is therefore

$$\frac{17!}{n!(17-n)!} \times \left[\frac{r_{\text{neutral}}}{(r_{\text{neutral}} + r_{\text{charged}})} \times f_{\text{neutral}} \right]^{17-n} \times \left[\frac{r_{\text{charged}}}{(r_{\text{neutral}} + r_{\text{charged}})} \times f_{\text{charged}} \right]^n$$

where r_{neutral} (=1.1) and r_{charged} (=0.9) are the reactivity ratios of neutral and charged POEGMA monomer, respectively, and each set of $(f_{\text{neutral}}, f_{\text{charged}}) = (0.93, 0.07), (0.77, 0.23),$ and $(0.64, 0.36)$ is the molar feed ratio of neutral and charged POEGMA monomer for synthesizing POEGMA7–PS, POEGMA23–PS, and POEGMA36–PS, respectively. To simplify the calculation, the degree of polymerization of POEGMA block was assumed to be 17.

Fig. R1 Composition distribution probability profiles of POEGMA#–PS (#: 7, 23, and 36).

As shown in **Fig. R1**, POEGMA7–PS, POEGMA23–PS, and POEGMA36–PS exhibit a probability of possessing uncharged diblock copolymer ($n = 0$) of 0.36, 0.024, and 0.0016, respectively. Since POEGMA7–PS has so few charged species along the backbone and a relatively small degree of polymerization of 17, it has a much greater probability of possessing a substantial amount of uncharged block copolymers than either POEGMA23–PS or POEGMA36–PS. Thus, as the targeting charge fraction decreases, it is more likely to be a mixture of uncharged and charged block copolymers. The inherent inhomogeneities in the charge distribution might play a significant role to the superlattice formation by inducing additional phase separation between charged and uncharged diblock copolymers. However, as POEGMA23–PS also displays a clear superlattice, the hypothesized segregation of charged and uncharged POEGMA blocks cannot be the major factor. In the revised manuscript, we have included Fig. R1 as a **Supplementary Fig. 8**, and also elaborated this valuable alternative hypothesis in the main text and Supplementary information as follows:

-Ln. 166: Furthermore, dispersity in the number of charges per chain could also play a significant role in superlattice formation. Since the POEGMA#–PS system possesses a relatively sparse distribution of charges along the low molecular weight POEGMA block, there might be inherent inhomogeneities in the charge distribution per each block copolymer, inducing a nontrivial phase separation between uncharged and charged species,¹⁹ which could contribute to the superlattice formation. A simple analysis (see Supplementary Fig. 8) suggests that for POEGMA7–PS, about one third of the POEGMA blocks could have no charged groups at all. However, for

POEGMA23–PS the fraction of uncharged chains drops below 3%, indicating that phase separation of charged and uncharged chains cannot be the primary driving force for superlattice formation.

ref 19) Sing, C. E. & Olvera de la Cruz, M. Polyelectrolyte blends and nontrivial behavior in effective Flory–Huggins parameters. *ACS Macro Lett.* **3**, 698–702 (2014).

-Supplementary Information (page 10, Supplementary Fig. 8): The component distribution probability profiles of POEGMA#–PS system can be estimated by assuming a simple binomial distribution of charged and uncharged POEGMA monomers along the backbone. The probability that the number of charged POEGMA monomers in POEGMA block equals n is

$$\frac{17!}{n!(17-n)!} \times \left[\frac{r_{\text{neutral}}}{r_{\text{neutral}} + r_{\text{charged}}} \times f_{\text{neutral}} \right]^{17-n} \times \left[\frac{r_{\text{charged}}}{r_{\text{neutral}} + r_{\text{charged}}} \times f_{\text{charged}} \right]^n$$

where r_{neutral} (=1.1) and, r_{charged} (=0.9) are the reactivity ratios of neutral and charged POEGMA monomer, respectively, and each set of $(f_{\text{neutral}}, f_{\text{charged}}) = (0.93, 0.07), (0.77, 0.23),$ and $(0.64, 0.36)$ is the molar feed ratio of neutral and charged POEGMA monomer for synthesizing POEGMA7–PS, POEGMA23–PS, and POEGMA36–PS, respectively. To simplify the calculation, the degree of polymerization of each POEGMA block was assumed to be 17.

As shown in Supplementary Fig. 8, POEGMA7–PS, POEGMA23–PS, and POEGMA36–PS exhibit a probability of possessing uncharged diblock copolymer ($n = 0$) of 0.36, 0.024, and 0.0016, respectively. Since POEGMA7–PS has only a few charged species along the backbone with a relatively small degree of polymerization of 17, it has a greater probability of possessing a substantial amount of uncharged block copolymers than POEGMA23–PS and POEGMA36–PS. Thus, as the targeting charge fraction decreases, it is more likely to be a mixture of uncharged and charged block copolymers.

Other minor comments:

-Ln. 130 - *The authors should specify that they consider χ_{eff} , or an effective χ , since the quantity they consider includes electrostatic interactions.*

Answer: We have incorporated the reviewer’s comment by specifying the effective χ as below:

-Ln. 130: ~since increasing the ion content drives up the effective value of χ .

-Ln. 153 - *I am unsure about the argument here. Yes, the Bjerrum length is long, but the presence of nearby charges (average distance $\ll l_B$ as mentioned in ln. 163) will cause any correlations - impactful as they should be to the thermodynamics of these systems - to ‘die down’ over a shorter length scale. I’ll recommend elaboration on this point, to clarify what the authors mean.*

Answer: The “mean-field” separation of ions on the order of about 2 nm is not, in fact, that much smaller than the Bjerrum length, so we do not necessarily agree that any short-range correlations will be suppressed. Furthermore, any net local attraction between charges and their counterions, such as those that lead to clustering in ionomers, would give rise to a larger length scale, on the same order of magnitude as both the domain spacing and Bjerrum length. The incorporation of this argument in the revised manuscript will be described below.

-Ln. 163 - Along these lines, if there are only a few charged monomers per chain, and even these short chains will interact with many neighboring other chains, in what way would they constrain the polymers be due to the electrostatic correlations? Would such an effect change the observation in ln. 130 that $d \sim \chi^{1/6}$? I am confused by the physical argument here, so further conceptual clarification or exposition would be helpful.

Answer: There are at least two effects in play here. First, as the average number of charges per chain increases, we expect that the effective χ between PS and POEGMA# will increase; hence the increase in d . Whether the length scale actually scales as $\chi^{1/6}$ we cannot yet say. Second, for modest amounts of charge, there is further structure formation, which we have vaguely (and perhaps imprecisely) attributed to “electrostatic correlations”. It might be more intuitive to view it as due to ionic clustering, analogous to the behavior of ionomers. The fact that the ions are tethered at the ends of relatively flexible side chains imbues them with significant conformational freedom, which could facilitate such clustering. Such clusters, if present, could preferentially lie in the middle of the POEGMA domains, or even conceivably at the interfaces. In the revision, we have further elaborated this point as follows:

-Ln. 164: \sim factor in driving formation of a superlattice. Furthermore, any net local attraction between charges and their counterions, such as those that lead to clustering in ionomers, and which could be facilitated by the conformational freedom of ions tethered at the ends of flexible POEGMA side chains, would give rise to a larger length scale, on the same order of magnitude as both the domain spacing and Bjerrum length.

-Ln. 178 - The strength of the charged interaction is due not only to the underlying partitioning, dielectric constant, and concentration, but also the size of the charge (i.e. the effective diameter of the species). This is true for both the Wang and Olvera de la Cruz predictions. While the use of a similar ion perhaps minimizes this as much as experimentally possible, the size of the pendant SO_3 versus the CF_3SO_3 may yet affect this comparison.

Answer: As the reviewer has commented, electrostatic interaction strength is also governed by the size of ionic species. In our experiments, however, it was necessary to use $NaCF_3SO_3$ salt despite the distinct anion sizes, since the $-CF_3$ moieties enhance the solubility of the salt in both organic solvent and POEGMA0–PS. In the revision, we have incorporated the reviewer’s comment as follows:

-Ln. 176: NaCF₃SO₃ partitions almost exclusively to the more polar POEGMA block due to the large solvation energy gained from the association of POEGMA and salt. Although a difference in anion size relative to the –SO₃ moieties in POEGMA#–PS could affect the electrostatic interaction strength, the overall effect on the enthalpic component of the effective χ parameter is assumed to be roughly equivalent in both cases.

Comments of Reviewer 3

This paper reports the synthesis and morphology of POEM-based polymers bearing charged moieties. The main finding of this work is the formation of interfacial layers from charge-containing block copolymers, which is unusual in literature. SAXS profiles in Fig. 2 are quite clear that the POEGMA7-PS and POEGMA23-PS form the superlattice. This reviewer thinks that this is an interesting paper to attract broad interests from polymer (physics, chemistry, and electrolytes) communities.

Questions remaining are (1) how those polymers can form so-called superlattice and (2) why not for POEGMA36-PS? In Fig. 3c and Fig. 3d, the authors speculated superlattice morphology. This is puzzling because of random copolymer characteristics of POEGMA, anticipated based on analogous reactivity ratios of two monomers. For POEGMA7-PS, the number of POEGMA-SO₃Na is 1.2. This implies that with nominally one POEGMA-SO₃Na unit, high electron density interfacial layers were developed to result in such sharp and multiple Bragg peaks. In fact, the scattering intensity of the peak at $q/(q^/2) = 1$ is strong, similar to that at $q/q^* = 1$. At the same time, the peak at $q/(q^*/2) = 1$ is quite sharp, compared with relatively broad peak at $q/q^* = 1$. To this reviewer, this is not easy to imagine how and why POEGMA-SO₃Na moieties randomly connected to neutral POEGMA are self-assembled into such structures. What would be the thickness of such layers? TEM image in Fig. 2 is not much informative.*

Answer: As the reviewer has described, charged POEGMA chains are randomly distributed with the neutral chains along the POEGMA backbone. However, the charged POEGMA chains should be strongly correlated, given that the Bjerrum length (l_B) of our system lies in the range of 6–11 nm, which is shorter than the superlattice length scale. Please note that the average spacing between charged species is estimated in POEGMA7–PS and POEGMA23–PS to be about 2.2 nm and 1.2 nm, respectively. As a consequence of such strong electrostatic cohesion between the charged species, POEGMA#–PS can self-assemble into a nanostructure. As noted in a previous response, the fact that the ions are tethered at the ends of relatively flexible side chains imbues them with significant conformational freedom, which could facilitate organization. Regarding the thickness of the charged POEGMA layer, it should be about 6 nm based on the SAXS profiles in Fig. 3a and 3b, but at this point we cannot be more specific.

With increased number density of tethered $-SO_3Na$ group, POEGMA23-PS having 3.9 units (that of neutral POEGMA is 13), the authors conjectured interconnected charged POEGMA phases, as shown in Fig. 3d. This reviewer believes that such speculation was made by the smeared superlattice peaks. But again, how randomly distributed charged POEGMA chains (1 per every 3-4 neutral units) can be assembled into such morphology?

Answer: For POEGMA23–PS the increased density of charges should facilitate adoption of a superlattice, while simultaneously possibly impeding the formation of very-well-defined subdomains.

One last minor point: If the sulfonate is in different forms, i.e., $-SO_3H$ or SO_3Li , does the sample show the same superlattice peaks? Or does it become indistinguishable because of low scattering contrast?

Answer: The reviewer has raised an important point on the effect of counterion properties on the morphologies. Although we have not experimentally elucidated this effect yet, alteration of counterion species should indeed tune the magnitude of the Coulomb potential and the strength of electrostatic cohesion, presumably leading to morphological changes. Sing *et al.* (ref 20; *Nat. Mater.* **2014**, *13*, 694–698) have also theoretically predicted such an effect on the phase behavior. Further studies on the morphological transition upon the alteration of the counterion species are currently in progress.

REVIEWERS' COMMENTS:

Reviewer #2 (Remarks to the Author):

The authors have adequately addressed all of my comments. I support publication of this work as-is in Nature Communications.

Reviewer #3 (Remarks to the Author):

The revised manuscript is ready for publication.